# Pivotal relationship between heavy metal, PM₂.₅ exposures and tuberculosis in Bangladeshi children: protocol paper of a case–control study

Rehnuma Haque [1,2] Molly Hanson,[2] Md Shariful Islam [3] Nazrin Akter,[1] Mohammad Moniruzzaman,[4] Md Jahangir Alam,[5] Md Kamruzzaman,[5] Mahbubur Rahman [1,2] Mohammod Jobayer Chisti [6] Rubhana Raqib,[7] Syed Moshfiqur Rahman[2]

For numbered affiliations see end of article.

**Correspondence to**
Dr Rehnuma Haque;
rehnuma.haque@uu.se

## ABSTRACT

**Introduction** Air pollution is a global issue that poses a significant threat to public health. Children, due to their developing physiology, are particularly susceptible to the inhalation of environmental pollutants. Exposure can trigger immune modulation and organ damage, increasing susceptibility to respiratory diseases. Therefore, we aim to examine the association between heavy metal and particulate matter exposure with tuberculosis in children.

**Methods and analysis** As a case–control study, we will include children diagnosed with pulmonary tuberculosis (n=60) and matched healthy controls (n=80) recruited from the same communities in Dhaka, Bangladesh. Exposure data for both cases and controls will be collected by a trained field team conducting home visits. They will administer an exposure questionnaire, measure child anthropometry, collect blood and household dust samples and instal 48-hour air quality monitors. The blood samples will be analysed by inductively coupled plasma mass spectrometry for serum heavy metal concentrations (lead, cadmium, arsenic, mercury and chromium), as a representative marker of exposure, and the presence of inflammatory biomarkers. Descriptive and inferential statistics, including independent samples t-tests, analysis of variance and conditional regression analysis, will be used to quantify heavy metal and particulate matter exposure status in tuberculosis cases compared with healthy controls, while accounting for potential confounders. Dust samples and air quality results will be analysed to understand household sources of heavy metal and particulate matter exposure. To test the study hypothesis, there is a positive association between exposure and tuberculosis diseases, we will also measure the accumulated effect of simultaneous exposures using Bayesian statistical modelling.

**Ethics and dissemination** This study has been approved by International Centre for Diarrhoeal Disease Research, Bangladesh's Institutional Review Board (PR-22030). The study findings will be disseminated at conferences and published in peer-reviewed journals.

## STRENGTHS AND LIMITATIONS OF THIS STUDY

⇒ Few case–control studies on a target population of children in highly polluted location.
⇒ Pilot study designed with a small sample size.
⇒ Extensive data collection for a variety of exposures and to control for confounders.
⇒ Unable to differentiate between other sources of exposure, including via food and drinking water.
⇒ Skewed blood metal concentrations will be log-transformed or categorised, potentially resulting in the loss of specific details.

## INTRODUCTION

Low-income and middle-income countries (LMICs) face the highest burden of both environmental pollution and respiratory infections. The WHO estimated globally more than one million children and young adolescents under 15 years fall ill with tuberculosis (TB) every year, and a quarter of those die.[1] An estimated 35 000 children with TB live in Bangladesh. However, determining the underlying causes and preventing TB in children are neglected as global health priorities and remain major challenges for Bangladesh.[2] Similar to many LMICs, Bangladesh is also overburdened by environmental toxicants and pollutants, both naturally occurring and exacerbated by anthropogenic activities. There is a growing evidence of high exposure to environmental pollutants in Bangladesh, through drinking water, soil, food, dust and ambient air.[3]

Vehicle emissions, spraying pesticides and fertilisers, industrial processes, mining, cooking and combustion are widespread anthropogenic sources of heavy metal pollutants, including arsenic (As), cadmium (Cd), lead (Pb), mercury (Hg) and chromium

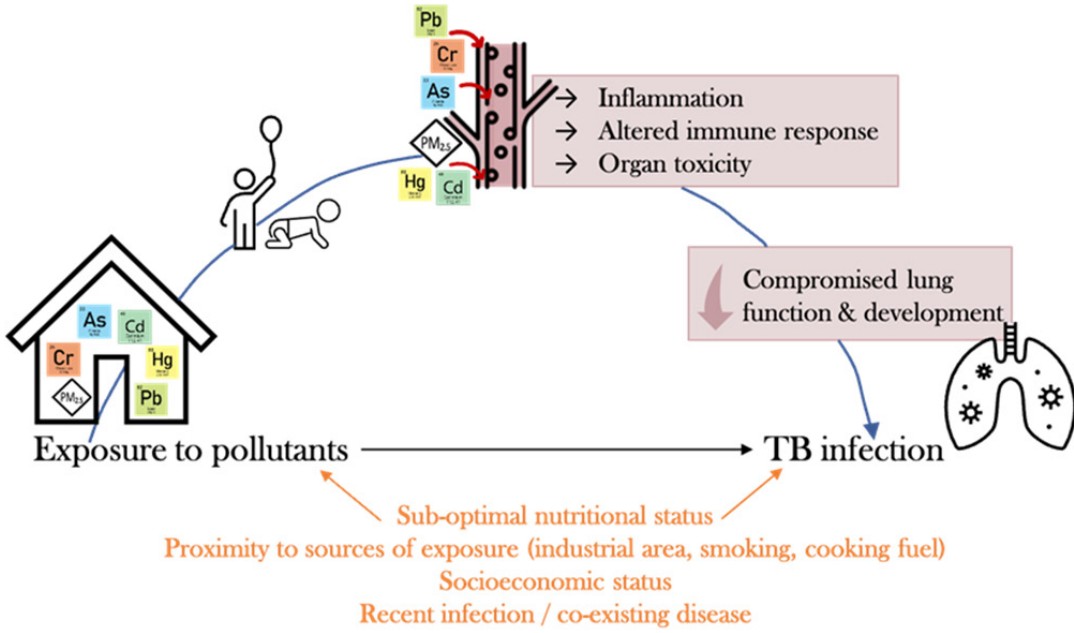

**Figure 1** Proposed causal pathway, adapted from existing Literature.[8] As, arsenic; Cd, cadmium; Cr, chromium; Hg, mercury; Pb, lead; TB, tuberculosis.

(Cr).[4 5] In Bangladesh, there are health concerns regarding the contamination of common food products and drinking water with heavy metals.[5] In addition, in 2020, AirVisual classified air quality in Bangladesh as 'unhealthy'.[6] Small airborne particle mixtures of heavy metals and organic species, such as carbon compounds, constitute respirable particulate matter (PM).[5 7] Living in Dhaka is a significant risk factor for exposure to air pollutants, as the capital city scores almost 20 times higher than the WHO recommended level of PM.[6 7] Therefore, through overlapping pathways of ingestion, inhalation and dermal absorption, exposure to potentially harmful levels of PM and heavy metals in Bangladesh is inevitable.

Airborne particles of heavy metals and PM cross from the respiratory system into the blood circulation.[5 7] The mechanisms of toxicity are not fully understood but contribute to inflammation, tissue damage and immune modulation, as described in figure 1.[8] Children are particularly susceptible to this environmental pollution toxicity due to their underdeveloped physiology and immature immune systems.[4 9] A growing body of research shows that children's lung growth and function is significantly worse in areas with higher air pollution.[4 9] Simultaneous exposure to multiple environmental pollutants and the interaction between heavy metals may also have synergistic effects.[8 10] Therefore, exposure to PM and heavy metals may alter the immune system, increasing susceptibility to pathogens and respiratory diseases, such as *Mycobacterium tuberculosis* bacteria.[9–11]

This paper describes the design of a patient-oriented pilot study which seeks to uncover the association between environmental heavy metals and TB. The rationale for the proposed study builds on existing research, highlighting that exposure to PM is significantly associated with an increased incidence of pulmonary TB (PTB) in adults.[12] However, the current body of evidence is limited on the environmental risks for children in Bangladesh. Furthermore, significantly greater progress is required to achieve the UN Sustainable Development Goal, target 3.3, to end TB epidemics by 2030.[13] We hypothesise that there is a positive association between PM and heavy metal exposure and PTB diseases in children. This study aims to bridge the current research gap, to understand the role of pollutants in TB pathogenesis, which is crucial to develop appropriate public health interventions and prevention activities.

We will achieve this with the following study objectives, to

1. Test the hypothesis that exposure to heavy metals (eg, Pb, As, Cd, Hg and Cr) and $PM_{2.5}$ is a risk factor for TB in children.
2. Quantify environmental sources of heavy metals, and PM exposure among children in urban areas in Dhaka city.

### Conceptual framework
Based on existing literature and evidence, this aim reflects a potential causal pathway (figure 1) connecting

inhalation of airborne pollutants with alterations in the immune system response. Toxic metals constitute ultra-fine ($PM_{2.5}$), considered the most harmful to the respiratory system, as the particles can deposit in the airways or cross the air–blood barrier in the lungs.[8 10] In the blood stream, Pb, As, Cd, Hg and Cr accumulate and have been shown to induce toxicity by generating reactive-oxygen species (ROS). ROS suppress the body's antioxidant defence, induce oxidative stress and alter the activation of specific proteins, lipids and enzymes involved in cell defence and the immune response.[8] Toxic metals disrupt immune homoeostasis, triggering excessive production of immunoglobulins (IgE and IgA) and proinflammatory cytokines and chemokines (TNF-α, IL-1β, IL-6), reduce numbers and/or impair function of immune cells (neutrophils, macrophages, natural killer cells, B and T lymphocytes and their subsets).[10 14] These overlapping pathways result in persisting inflammation, immune modulation and organ toxicity.

TB pathogenesis can be attributed to increased susceptibility to infectious diseases and impairment in lung function.[8 10] Due to their underdeveloped physiology, children's immature lungs are uniquely susceptible to respiratory tract infections.[4 10] Children are crawling, playing and breathing in the air closest to the ground, where high levels of pollutants are concentrated.[4] In contrast to adults, children under 5 have a faster breathing rate and a relatively immature immune system, making them less efficient in responding to pollutants and pathogens.[9]

## METHODS

### Study design and setting

We will conduct a case–control study to compare exposure to heavy metals among children diagnosed with TB (case) and matched healthy controls in different urban areas of Dhaka City, Bangladesh.

The population of Dhaka has multiplied over four times since 1980, attaining one of the highest population densities in the world, with a remarkable 24 million inhabitants. With persisting urbanisation attracting people to the city, this number is expected to continue growing. As a consequence, rapid urbanisation has aggravated the concentration of pollutants in air, soil and water.[15] Bangladesh, and especially the capital city of Dhaka, has recorded some of the worst air pollution in the world.[6] Existing studies suggest wide-ranging health consequences from exposure to heavy metals and PM through ingestion, inhalation and dermal absorption.[3 5 7] Bangladesh was selected as the study site as it shares one of the highest burdens of TB diseases in the world.[2] The study is hosted by the Environmental Health and WASH group at icddr,b.

We will measure whole blood for heavy metal concentrations to assess exposure and conduct immunological assays to evaluate immune status of the patient. We will also conduct household air and dust sampling to identify some sources of heavy metal exposure among children. Household geo-mapping, an exposure questionnaire and

socioeconomic surveys from the parents, will be used to further explore the sources of metal exposure among the children. Written consent will be obtained from guardians/parents and assent from older children (10–14 years) before the data and specimen collection.

### Study participants

We will recruit children between the ages of 1 and 14 living in Dhaka city who have been diagnosed with PTB based on the Bangladesh national TB guideline recommended approach by a registered physician.[16] The national TB guideline recommended a comprehensive approach which includes a careful history taking to identify potential exposure and symptoms suggestive of PTB, a clinical assessment that involves weight monitoring and chest X-rays, bacteriological confirmation through tests like smear microscopy and culture, and additional supportive investigations as needed.

At the beginning of the project, we will visit the TB unit of Dhaka Shishu (Children) Hospital, which is specifically designated for paediatric TB patient's ongoing diagnosis and treatment. To identify eligible participants, we will obtain a registry of TB-diagnosed children, providing the names, ages, addresses and contact details of children diagnosed with PTB. We will then contact the parents or caregivers to request for their child's participation in our study.

Following our inclusion and exclusion criteria, we will enrol 50 children diagnosed with PTB. In addition, we will then select 50 healthy controls between the ages of 1 and 14 years old who live in the same communities as the cases. According to Rothman et al, it is important to match cases with the controls from similar geographic locations as a partial adjustment of environmental and socioeconomic factors.[17] The process of selecting controls will be carried out carefully to minimise bias and ensure the validity of our study results. We will identify potential controls by conducting a door-to-door survey within a 0.5 km radius of each case child's residence. We will select the first subsequent child of the same age who participates in shared activities, such as schooling, playing, learning or mosque visits. To ensure the control children are healthy, the exclusion criteria for controls will be: (1) hospitalisation in the past 60 days and/or discharged from hospital in the past 30 days; (2) known chronic disease; (3) very sick appearance requiring medical attention; (4) recent cough with blood or painful ear infections or (5) recovery from diagnosed TB.

To account for potential drop-outs, we aim to enrol a total of 60 children in the case group and 80 children in the control group.

### Sample size calculation

To determine our required sample size, we have considered the sample size for similar case–control studies with continuous exposures, as described by Lubin et al.[18] The formula used to calculate sample size was:

$$n = \left(\frac{k+1}{k}\right)\frac{\bar{p}\left(1-\bar{p}\right)\left(z_\beta+z_{\alpha/2}\right)^2}{\left(p_1-p_2\right)^2}$$

where,
k=ratio of control to case, $p_1$=proportion of cases exposed, $p_2$=proportion of controls exposed.
ß = power.
α=level of significance.

$$\bar{p} = \frac{\left(p_1+p_2\right)}{2}$$

We estimate that a sample size of 50 children, in both the case and control groups, would achieve 80% power to detect a mean difference in blood concentrations and OR of 5 at a significance level of 0.05. Recruiting 60 children in the case group and 80 children would allow for a drop-out of 10 and 20 children in each group, respectively.

### Exposure questionnaire and anthropometry

Over a period of 6–8 months, a trained field team will conduct home visits and administer a questionnaire to the mother or primary caregiver on household exposures. The questionnaire will cover a range of topics including socioeconomic status, demographics, sources of environmental and household pollution (such as smoking, type of cooking fuel, proximity to battery/textile/waste incineration areas), and other behaviours that may impact pollutant exposure (such as engagement in industrial manufacturing or agricultural practices). Additionally, we will inquire about the children's medical history, including any recent infections or coexisting conditions.

An experienced medical technologist will also gather anthropometric data to assess the child's overall nutritional status. The child's height and weight will be measured using a standardised wooden height scale and digital weight scale (Beurer ps240). These will be validated for accuracy and precision at icddr,b before going to the field.

Clear instructions will be provided to the child prior to weighing, emphasising the importance of removing any items that could affect the measurement, such as shoes or heavy jackets. The child will be positioned with their weight evenly distributed on both feet, and they will be asked to stand still on the weighing machine with their arms at their sides. The weight measurement will be taken in kilograms and repeated. If the two measurements are noticeably different, we will repeat the process and calculate the average of three measurements to ensure accuracy. Similarly, when measuring height, the child will remove any items that may affect the measurement, such as shoes, socks, headgear or hair accessories. The child will be informed to stand upright and look straight ahead. Measurements will be taken in centimetres. For infants under 2 years of age, or who cannot stand, we will adjust the anthropometric measurements by laying them on a flat surface for height and in a weighing tray for weight.

### Biological sample collection

During the household visits, the field team will also collect blood samples, household dust and indoor air samples, from both case and control groups. These samples will be analysed to compare biomarkers and environmental exposures between the two groups. By characterising the sources and risk of pollutant exposure, we can propose intervention strategies to reduce the burden of paediatric TB in LMICs.

To collect the blood sample, the area around the cubital vein will first be sterilised. We will use a S-Monovette Metal analysis Lithium-Heparin 7.5 mL tube and Safety-Multifly needle to draw 5 mL of blood from the child's vein. Each sample tube will be carefully labelled with a unique barcode and anonymous relevant information, including the date of collection and household ID. After collecting the blood sample tubes will be transported in cool box to the Immunobiology, Nutrition and Toxicology Laboratory at icddr,b for analysis. A 1 mL of whole blood will be removed into cryovials, and the remaining blood will be centrifuged to separate plasma from red blood cell components. All aliquots will be stored at −80˚C.

For heavy metal concentrations in blood/plasma, the primary exposures of interest are Pb and Cd, while secondary exposures include As, Hg and Cr. Aliquots of blood will be analysed for these metals by inductively coupled plasma mass spectrometry (ICP-MS). As we will conduct ICP-MS analysis; therefore, we will get the panel of heavy metal results at once. The immunoglobulins, proinflammatory cytokines and chemokines will be tested in plasma by ELISA or automated immune analyser as appropriate. The leftover samples (whole blood and plasma) will be preserved for future analysis of inflammatory and nutritional markers in relation to environmental exposures mostly heavy metals complying with the icddr,b specimen storage policy, for relevant analysis depending on the future availability of funds.

### Environmental sample collection

We will instal portable air quality monitors (PurpleAir) in participants' homes to measure real-time $PM_{2.5}$ levels for at least 48 hours. These will be placed in the child's bedroom or other places where they spend most of their time at home. Data on $PM_{2.5}$, humidity and temperature will be sent directly to our smart devices.

During the household visits, the team will collect samples of household dust from the children's homes. These will be analysed at the, icddr,b using X-ray fluorescence spectrometry (XRF). A subset of dust samples will be sent to the Bangladesh Council of Scientific and Industrial Research laboratory for quality assurance, where repeat measures will be performed by ICP-MS and data will be compared with XRF data. The results will be used to fulfil the second research objective, to identify and quantify environmental sources of heavy metals and PM.

## Data management and analysis plan

A data management standard operating protocol will be developed considering the range and variation in data types. Field researcher officers will consult with the research assistants on a daily basis to check data consistency. The household questionnaire data will be digitally programmed through tablet computers, downloaded from the devices each week and reviewed by a statistician. The statistician will update the server and produce datasets for the investigators in R program language. Once the data are available to the investigators, they will run previously developed Stata do files to assess inconsistencies and send queries to the field team. All metadata will be stored on the icddr,b's Data Repository System and a backup will be taken.

As evidenced in similar studies, the distributions of Pb, As, Cd, Hg and Cr in the blood samples are expected to be skewed. In order to fulfil the assumptions of the selected parametric tests, this data will be transformed, by either natural log transformation or categorisation of the metal concentrations.

Descriptive analysis, including frequency tests, will be stratified by child's age, sex and TB condition. $\chi^2$, independent samples t-tests or analysis of variance will be performed to compare the blood heavy metal concentrations among TB cases and healthy controls. In addition, we will compare the sociodemographic information, based on the parental exposure questionnaire survey. We will use conditional regression analysis to estimate the biomarkers that are associated with heavy metal and PM exposures among TB children. In addition, if we identify two or more elevated metals in the blood and/or environmental samples, we will measure their accumulated effect using Bayesian statistical modelling.[19]

## ETHICS AND DISSEMINATION

The study design and protocols have received approval from the Institutional Review Board (PR-22030) of the International Centre for Diarrhoeal Disease Research, Bangladesh (icddrb). This board comprises the Research Review Committee (RRC) and the Ethical Review Committee (ERC). Field activities and sample collection for the study commenced on 1 June 2023 and will continue until 25 May 2024.

There are minor risks to the participants during sample collection. The risks and advantages for participants will be provided in written information forms to ensure the participants are fully informed about the study. The purpose and methods of the study will also be clearly described. This information will be written in English and translated into Bangla in a format that is easy to understand even with little or no educational background. If required, the form will also be read out to the participant.

As this study recruits children between 1 and 15 years of age, the mothers or caregivers will be required to give consent on behalf of their children participating in the study. Before taking their consent, the researchers will repeat the study information and participants will have time to ask any questions. The participants will also be informed that they have the freedom to withdraw from the study at any time and have no adverse implications. Those who agree to participate will be required to give written informed consent, by signing or making an imprint of their left thumb and verified by an independent witness. Participants will also keep a copy of the information and consent forms so they have open access and can contact the research review committee if necessary.

Field data and the sample test results will be uploaded to the secured server at icddr,b. Only investigators will have access to the dataset. Prior to uploading the data, the field supervisor will check the datasets for consistency and identify any missing information. If any inconsistencies are identified, queries will be sent to the field and resolution of the discrepancies will be made. Throughout the data management, confidentiality of information will be strictly maintained, and access to the data will be restricted to the research team members only. At the time of analysing information and publishing the results of the study, the name or identity of the study participants will not be used.

The study results will be shared within icddrb through scientific seminars and the annual report. Additionally, we will disseminate the findings to our collaborators and participate in national and international conferences. Also, the results will be published in peer-reviewed journals

## Patient and public involvement

Patients were not involved to develop this protocol paper.

## Expected results

The hypothesis-driven research we propose is intended to address the gap in developing childhood TB diseases and the association with environmental pollutant exposure, with a focus on air pollution and heavy metal exposure. This research will provide evidence for follow-up studies to improve understanding of the causal pathway connecting these pollutants to TB diseases and inform evidence-based guidance to eliminate TB. Our long-term goal is to evaluate the chemical risk factors for respiratory infections and to improve children's health.

### Author affiliations

[1]Environmental Health and WASH, Health System Population and Studies Division, International Centre for Diarrhoeal Disease Research, Mohakhali, Dhaka, Bangladesh
[2]Global Health and Migration Unit, Department of Women's and Children's Health, Uppsala University, Uppsala, Sweden
[3]The Department of Public Health, The University of Queensland, Brisbane, Queensland, Australia
[4]Bangladesh Council of Scientific and Industrial Research, Dhaka, Bangladesh
[5]Dhaka Shishu Hospital, Dhaka, Bangladesh
[6]Nutrition Research Division, International Centre for Diarrhoeal Disease Research, Mohakhali, Dhaka, Bangladesh
[7]Immunobiology Nutrition and Toxicology Laboratory, Nutrition Research Division, International Centre for Diarrhoeal Disease Research, Dhaka, Bangladesh

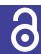

𝕏 Rehnuma Haque @Rehnuma_H

**Contributors**  RH conceptualised the study, drafted the original manuscript and secured the funding. MH, MSI, and SMR supported in the first draft of manuscript development. RH, NA, MR, MJC, RR, MJA and MK provided input in study design and site selection. RR and MM provided support in laboratory SOP development. All authors participated in editing, reviewing and approved the final version of the manuscript for publication.

**Funding**  This research is funded by Thrasher Research Fund- Early Career Award number 01477 and Uppsala University, Sweden.

**Competing interests**  None declared.

**Patient and public involvement**  Patients and/or the public were not involved in the design, or conduct, or reporting, or dissemination plans of this research.

**Patient consent for publication**  Not applicable.

**Provenance and peer review**  Not commissioned; externally peer reviewed.

**ORCID iDs**
Rehnuma Haque http://orcid.org/0000-0002-4712-7709
Md Shariful Islam http://orcid.org/0000-0002-3015-0830
Mahbubur Rahman http://orcid.org/0000-0003-0520-2683
Mohammod Jobayer Chisti http://orcid.org/0000-0001-9958-3071

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
