## [Reviewer comments · BMJ Open]

ARTICLE DETAILS

TITLE (PROVISIONAL)	The pivotal relationship between heavy metal exposures and tuberculosis in Bangladeshi children: Protocol paper of a case-control study
AUTHORS	Haque, Rehnuma; Hanson, Molly; Shariful Islam, Md; Akter, Nazrin; Moniruzzaman, Mohammad; Alam, Md. Jahangir; Kamruzzaman, Md.; Rahman, Mahbubur; Chisti, Md.; Raqib, Rubhana; Rahman, Syed

VERSION 1 – REVIEW

REVIEWER	Sarkar, Shamim International Centre for Diarrhoeal Disease Research Bangladesh
REVIEW RETURNED	07-Aug-2023

GENERAL COMMENTS	Overall comments: Rehnuma Haque Sarah (bmjopen-2023-075010) and colleagues developed a protocol to examine the relationship between heavy metal exposures and tuberculosis in Bangladeshi children using a case-control study. The topic is interesting to generate preliminary evidence regarding the relationship between heavy metal exposure and the incidence of tuberculosis in children critical in mitigating tuberculosis risk among the heavy metal-exposed population. The overall study is well written, and I have some comments below for consideration: Comment 1: The author mentions in lines 14-15 that this paper describes the design of a patient-oriented pilot study that seeks to uncover the interplay and potential causal pathway between environmental heavy metals and tuberculosis. Does a case-control study establish the temporal relationship between exposure and outcome? Please rephrase these sentences to reflect the application of a case-control study in your research. Comment 2: In lines 15-16, please provide a justification for enrolling children aged between 1 and 14 years in your study. Comment 3: What are the statistical justifications for opting for a 1:1 ratio in this case-control study? Why was a ratio of 1:2 or 1:3 not chosen, as it could potentially enhance the study's statistical power? Comment 4: In sample size calculation section: Why is there no information about the expected proportion of exposure to heavy metals in both the control and case groups? Why was an assumed odds ratio of
---

	5 chosen? What is the rationale behind this odds ratio, and is there a reference to support it? Comments 5: In analysis plan section: Why did the author not consider utilizing conditional logistic regression to analyze the matched case-control data in this study? Instead, the author opted for applying Generalized Additive Modeling (GAM) in analyzing the matched case-control data in this study. To ensure clarity for readers who may not have an extensive statistical background, it is advisable to include a succinct explanation of why Generalized Additive Modeling (GAM) was chosen over conditional logistic regression in the Materials and Methods section. This explanation should be 1 or 2 sentences long and easy to understand for readers without a statistical background.
--	---

REVIEWER	Chakaya, Jeremiah Kenyatta University/Liverpool University
REVIEW RETURNED	14-Aug-2023

GENERAL COMMENTS	This paper describes a protocol of a proposed case control study intended to examine the relationship between heavy metal exposure and exposure to particulate matter (the exposure variables) with TB disease (the outcome). The study team proposes to recruit 50 children diagnosed to have TB (the cases) and 80 children who are “healthy” (the controls) and in both groups measure levels of heavy metals (lead, cadmium, arsenic, mercury and chromium) and assess air pollution levels in the dwellings of these children. The authors expect that this exploratory study will provide information that will be useful in the design of programs for the prevention of TB in children living in low- and middle-income countries where the burden of childhood TB is high. While the study is interesting and should be implemented , I would like to raise a number of issues that the authors should consider.  1. There seems to be a conflation of TB Infection and TB disease. The authors appear to be interested in TB disease as the outcome variable but are repeatedly using TB infection in the description of this outcome. While all people with TB disease are infected with Mycobacterium tuberculosis, the large majority of people with Mycobacterium tuberculosis infection (Tuberculosis Infection or TBI) in the world today, do not have disease and a large proportion of them will not develop TB disease in their lifetime. If the authors are also interested in examining TB infection in addition to TB disease, this needs to be clearly described in the protocol and the test that will be used to diagnose TBI defined (the tuberculin skin test or the interferon gamma release assay). 2. The assessment of environmental sources of heavy metals will be based primarily on household air and dust sampling and not ambient (outdoor) air sampling. This is likely to limit the achievement of objective 2 , the assessment of environmental sources of pollutants and thus introduce bias. Will proximity to a major road be included among the environmental sources of pollution for example? 3. The protocol proposes to assess the association between heavy metal exposure and the production of pro-inflammatory cytokines. The cytokines of interest are not listed and on lines 47-52, there is another conflation between cellular markers of inflammation (such as CD8+ and NK cells) and inflammatory cytokines.
---

	4. It appears that cases will be primarily children with pulmonary TB. In children extra-pulmonary forms of TB are common and the younger the child the more common extra-pulmonary (and often severer TB) is. Will these children be excluded? The definition of pulmonary TB provided by the authors appears to be erroneous. It is stated in the paper that these children will have bacteriological confirmation of TB and without radiographic abnormalities. Pulmonary TB without radiographic abnormalities is rare. 5. Healthy children , as per the protocol will be children who have not been hospitalized, suffered a cough or a recent infection including painful ear infection. It is known that being asymptomatic does not necessarily mean absence of TB disease. Will the control children be chest x-rayed in addition to the administration of a symptom questionnaire and obtaining anthropometric measurement? The chest x-ray is a more sensitive tool for TB screening and mostly when normal, it excludes the presence of active TB. 6. It is noted that blood samples from consenting children will be banked for future studies , however , the nature of these studies is unclear. It is important that consenting parents/assenting children are fully aware of the studies to be carried out in the future including the biomarkers that will be measured. 7. It is not clear why arsenic, mercury and chromium are considered secondary exposures as opposed to lead and cadmium. 8. This is a speculative statement but also an important one to consider because it has implications for the interpretation of the data that will be collected. It is noted that cases and controls will be from the same “environment” or “community” , which may mean they suffer similar levels of pollution. In essence then the value of the study may in fact be in providing evidence that children with TB handle these exposures differently from those who do not develop TB.
--	---

VERSION 1 – AUTHOR RESPONSE

Reviewer: 1

Dr. Shamim Sarkar, International Centre for Diarrhoeal Disease Research Bangladesh

Comments to the Author:

Overall comments:

Rehnuma Haque Sarah (bmjopen-2023-075010) and colleagues developed a protocol to examine the relationship between heavy metal exposures and tuberculosis in Bangladeshi children using a case-control study. The topic is interesting to generate preliminary evidence regarding the relationship between heavy metal exposure and the incidence of tuberculosis in children critical in mitigating tuberculosis risk among the heavy metal-exposed population. The overall study is well written, and I have some comments below for consideration:

Comment 1:

The author mentions in lines 14-15 that this paper describes the design of a patient-oriented pilot study that seeks to uncover the interplay and potential causal pathway between environmental heavy metals and tuberculosis.

Does a case-control study establish the temporal relationship between exposure and outcome?
Please rephrase these sentences to reflect the application of a case-control study in your research.
Response: Many thanks for raising this important point. No, case-control study cannot establish a temporal relationship between exposure and outcome. We have revised the sentence as follows in Page 3, L100-101.

“the design of a patient-oriented pilot study which seeks to uncover the association between environmental heavy metals and tuberculosis.”

Comment 2:

In lines 15-16, please provide a justification for enrolling children aged between 1 and 14 years in your study.

Response: TB disease in children under 15 years of age is called paediatric tuberculosis, according to the WHO Consolidated Guidelines on Tuberculosis: Module 5: Management of Tuberculosis in Children and Adolescents 2022 and NTP. 2021. 'National Guideline and Operational Manual for Tuberculosis, Tuberculosis Control Programme (NTP), Directorate General of Health Services Ministry of Health and Family Welfare, Dhaka, Bangladesh. Therefore, we have selected 1-14 years age range. We put reference in page 10, L437-439 and L458-460.

Comment 3:

What are the statistical justifications for opting for a 1:1 ratio in this case-control study? Why was a ratio of 1:2 or 1:3 not chosen, as it could potentially enhance the study's statistical power?

Response: We thank the reviewer for the comment. We used 1:1 ratio because tuberculosis in children is relatively common, and the study aims to assess the association between heavy metals and tuberculosis.

Comment 4:

In sample size calculation section: Why is there no information about the expected proportion of exposure to heavy metals in both the control and case groups? Why was an assumed odds ratio of 5 chosen? What is the rationale behind this odds ratio, and is there a reference to support it?

Response: Thank you for raising this very important question. Given that this is a pilot study, we did not conduct a sample size calculation specific to the odds ratio. We also consider this odds ratio to limit the sample size within 100 participants.

Comments 5:

In analysis plan section: Why did the author not consider utilizing conditional logistic regression to analyze the matched case-control data in this study? Instead, the author opted for applying Generalized Additive Modeling (GAM) in analyzing the matched case-control data in this study. To ensure clarity for readers who may not have an extensive statistical background, it is advisable to include a succinct explanation of why Generalized Additive Modeling (GAM) was chosen over conditional logistic regression in the Materials and Methods section. This explanation should be 1 or 2 sentences long and easy to understand for readers without a statistical background.

Response: Thank you for noticing this important analysis error. We have removed GAM from our analysis plan and have included conditional regression analysis, as per reviewer's suggestion. We understand that in a matched case-control study the analysis plan should use a conditional regression model. Kindly see page 1, L37 and page 8, L335.

Reviewer: 2

Dr. Jeremiah Chakaya, Kenyatta University/Liverpool University

Comments to the Author:

This paper describes a protocol of a proposed case control study intended to examine the relationship between heavy metal exposure and exposure to particulate matter (the exposure variables) with TB disease (the outcome). The study team proposes to recruit 50 children diagnosed to have TB (the

cases) and 80 children who are “healthy” (the controls) and in both groups measure levels of heavy metals (lead, cadmium, arsenic, mercury and chromium) and assess air pollution levels in the dwellings of these children. The authors expect that this exploratory study will provide information that will be useful in the design of programs for the prevention of TB in children living in low- and middle-income countries where the burden of childhood TB is high. While the study is interesting and should be implemented, I would like to raise a number of issues that the authors should consider.

1. There seems to be a conflation of TB Infection and TB disease. The authors appear to be interested in TB disease as the outcome variable but are repeatedly using TB infection in the description of this outcome. While all people with TB disease are infected with *Mycobacterium tuberculosis*, the large majority of people with *Mycobacterium tuberculosis* infection (Tuberculosis Infection or TBI) in the world today, do not have disease and a large proportion of them will not develop TB disease in their lifetime. If the authors are also interested in examining TB infection in addition to TB disease, this needs to be clearly described in the protocol and the test that will be used to diagnose TBI defined (the tuberculin skin test or the interferon gamma release assay).

Response: We thank the reviewer for noticing this inconsistency. Our research aim is to understand the impact of environmental exposure on ‘TB diseases’ and not on ‘TB infection’. We have revised this throughout the protocol paper. Kindly see pages: Page 1, L41; page 3, L108; page 4, L179; page 9, L375 and L378.

2. The assessment of environmental sources of heavy metals will be based primarily on household air and dust sampling and not ambient (outdoor) air sampling. This is likely to limit the achievement of objective 2, the assessment of environmental sources of pollutants and thus introduce bias. Will proximity to a major road be included among the environmental sources of pollution for example?

Response: We thank the reviewer for a very useful comment. We also agree that ambient air pollution can indeed influence indoor PM levels. Particularly we will focus on the indoor environment in this project. Therefore, we will collect indoor air PM data and household dust sample. We do have the option to utilize real-time data available on our icddr website at <https://cch.icddr.org/> which includes hourly records of PM_{2.5}, PM₁₀, and PM₁ levels, humidity, and temperature. If necessary especially during the data analysis phase we can use this data for comparisons between ambient and indoor air pollution. Additionally, our exposure questionnaire encompassed various factors, including household proximity to roads, nearby any industrial activities, cooking fuel type, indoor smoking etc.

3. The protocol proposes to assess the association between heavy metal exposure and the production of pro-inflammatory cytokines. The cytokines of interest are not listed and on lines 47-52, there is another conflation between cellular markers of inflammation (such as CD8+ and NK cells) and inflammatory cytokines.

Response: Many thanks to the reviewer for bringing our attention to this important point. As indicated, we have revised the section in page 3, L128-132 and page 7, L293-295 and have also included the names of some potential cytokines that will be determined in our study with a citation as follows: “Page 3, L128-132: Toxic metals disrupt immune homeostasis, triggering excessive production of immunoglobulins (IgE and IgA), and pro-inflammatory cytokines and chemokines (TNF- α , IL-1 β , IL-6), may reduce numbers and/or impair function of immune cells (neutrophils, macrophages, natural killer cells, B and T lymphocytes, reduce memory B and T lymphocytes and their subsets) (Anka et al. 2022; Ackland et al. 2015).

Page 7, L293-295: The immunoglobulins, pro-inflammatory cytokines and chemokines will be tested in plasma by enzyme-linked immunosorbent assay (ELISA) or automated immune analyser as appropriate.

Page 9, L403-407: Anka, Abubakar U, Abubakar B Usman, Abubakar N Kaoje, Ramadan M Kabir, Aliyu Bala, Mandana Kazem Arki, Nikoo Hossein-Khannazer, and Gholamreza Azizi. 2022. ‘Potential Mechanisms of Some Selected Heavy Metals in the Induction of Inflammation and Autoimmunity’.

European Journal of Inflammation 20 (September): 1721727X221122719.
[https://doi.org/10.1177/1721727X221122719.](https://doi.org/10.1177/1721727X221122719)”

4. It appears that cases will be primarily children with pulmonary TB. In children extra-pulmonary forms of TB are common and the younger the child the more common extra-pulmonary (and often severer TB) is. Will these children be excluded? The definition of pulmonary TB provided by the authors appears to be erroneous. It is stated in the paper that these children will have bacteriological confirmation of TB and without radiographic abnormalities. Pulmonary TB without radiographic abnormalities is rare.

Response: We appreciate the comment by the reviewer. We have revised the recruitment statement in page 5, L193-199 and L202-203.. We will recruit each child already diagnosed with PTB from our collaborating partner Dhaka Shishu (Children) Hospital registry. This hospital has specialized TB care unit for the children and follow Bangladesh national Tuberculosis diagnosis and management guidelines for diagnosis and treatment.

Recommendation of diagnosis criteria as follow:

1. Careful history taking (including history of contact with TB and symptoms suggestive of TB)
2. Clinical assessment (including serial weight monitoring/growth assessment)
 - Chest X-ray and other radiological evaluation
 - Bacteriological confirmation whenever possible: Smear microscopy, Xpert-MTB RIF, Culture of respiratory sample / gastric lavage
 - Other supportive investigations relevant to suspected PTB/EPTB

Reference: [https://www.ntp.gov.bd/wp-content/uploads/2021/10/Operational-Manual-for-Tuberculosis_compressed.pdf.](https://www.ntp.gov.bd/wp-content/uploads/2021/10/Operational-Manual-for-Tuberculosis_compressed.pdf))

- HIV testing

We are will exclude extrapulmonary TB children in this project.

5. Healthy children , as per the protocol will be children who have not been hospitalized, suffered a cough or a recent infection including painful ear infection. It is known that being asymptomatic does not necessarily mean absence of TB disease. Will the control children be chest x-rayed in addition to the administration of a symptom questionnaire and obtaining anthropometric measurement? The chest x-ray is a more sensitive tool for TB screening and mostly when normal, it excludes the presence of active TB.

Response: Thank you for your comment. Unfortunately, healthy children have not been screened for the chest X-ray as we recruited healthy children from the community by door-to-door visit and we are not equipped with portable X-ray machine. However, we will collect anthropometric measurement and blood sample for complete blood count (CBC) to see if there is any sign of infections present or not. Also, questionnaire-based data will be collected to assess health/morbidity status such as cold, cough, fever etc.in the last 1-2 months.

6. It is noted that blood samples from consenting children will be banked for future studies , however , the nature of these studies is unclear. It is important that consenting parents/assenting children are fully aware of the studies to be carried out in the future including the biomarkers that will be measured.

Response: We have obtained consent for the future utilization of the biological samples mentioned in page 5, L188-189.

7. It is not clear why arsenic, mercury and chromium are considered secondary exposures as opposed to lead and cadmium.

Response: Thanks for raising your concern. As we will conduct ICP-MS analysis therefore we will get the panel of heavy metal result at once.: We have had to prioritize the Pb and Cd as substantial evidence of Pb and Cd contamination among children in Dhaka, which is significantly higher found in-

Chowdhury, K.I.A., Nurunnahar, S., Kabir, M.L., Islam, M.T., Baker, M., Islam, M.S., Rahman, M., Hasan, M.A., Sikder, A., Kwong, L.H., Binkhorst, G.K., Nash, E., Keith, J., McCartor, A., Luby, S.P., Forsyth, J.E., 2021. Child lead exposure near abandoned lead acid battery recycling sites in a residential community in Bangladesh: Risk factors and the impact of soil remediation on blood lead levels. *Environ. Res.* 194, 110689. <https://doi.org/10.1016/j.envres.2020.110689>

Nargis, A., Habib, A., Islam, M.N., Chen, K., Sarker, M.S.I., Al-Razee, A.N.M., Liu, W., Liu, G., Cai, M., 2022. Source identification, contamination status and health risk assessment of heavy metals from road dusts in Dhaka, Bangladesh. *J. Environ. Sci.* 121, 159–174. <https://doi.org/10.1016/j.jes.2021.09.011>

In rural Bangladesh, arsenic poses a greater risk primarily because of the prevalent use of tube well water. However, we acknowledge that the other two metals of interest, Cr and Hg, are also assumed to contribute significantly to the environmental exposure, particularly given the high presence of these metals in poultry feed.

8. This is a speculative statement but also an important one to consider because it has implications for the interpretation of the data that will be collected. It is noted that cases and controls will be from the same “environment” or “community”, which may mean they suffer similar levels of pollution. In essence then the value of the study may in fact be in providing evidence that children with TB handle these exposures differently from those who do not develop TB.

Response: Thank you for your comment. We also have a hypothesis that factors other than pollution levels may contribute to a child's vulnerability to TB, even when exposed to the same level of polluted environment. To explore this, we planned to investigate their blood for heavy metal concentrations, various immunomarkers, and other sociodemographic factors to better understand the impact of pollution.

Reviewer: 1

Competing interests of Reviewer: I have declared that no competing interests exist.

Reviewer: 2

Competing interests of Reviewer: I have no competing interests

VERSION 2 – REVIEW

REVIEWER	Sarkar, Shamim International Centre for Diarrhoeal Disease Research Bangladesh
REVIEW RETURNED	13-Nov-2023

GENERAL COMMENTS	The authors have addressed the comments and suggestions. I do not have any further comments.
--

REVIEWER	Chakaya, Jeremiah Kenyatta University/Liverpool University
REVIEW RETURNED	24-Nov-2023

GENERAL COMMENTS	The revised version of this manuscript addressed the issues raised in the previous review satisfactorily. I have no further comments/inputs.
--

VERSION 2 – AUTHOR RESPONSE